# The Impact of Assessment of Nurses’ Experiences in Thoracic Surgery in Onco-Hematological Patients

**DOI:** 10.3390/healthcare12181843

**Published:** 2024-09-14

**Authors:** Gaetana Messina, Giovanni Natale, Caterina Sagnelli, Giovanni Vicidomini, Diana Mancino, Giuseppe Cerullo, Simona De Gregorio, Sabrina De Angelis, Carmela Otranto, Beatrice Leonardi, Silvia Dattolo, Noemi Maria Giorgiano, Andrea De Masi, Francesco Esposito, Maria Antonietta Puca, Giuseppe Vicario, Alfonso Fiorelli, Antonello Sica

**Affiliations:** 1Thoracic Surgery Unit, Azienda Ospedaliera Universitaria “Luigi Vanvitelli”, University of Campania “Luigi Vanvitelli”, 80131 Naples, Italy; adamessina@virgilio.it (G.M.); giovanni.natale@unicampania.it (G.N.); giovanni.vicidomini@unicampania.it (G.V.); diana.mancino@unicampania.it (D.M.); giuseppe.cerullo@unicampania.it (G.C.); simona.degregorio@policliniconapoli.it (S.D.G.); sabrina.deangelis@policliniconapoli.it (S.D.A.); ort.carm@libero.it (C.O.); beatrice.leonardi@unicampania.it (B.L.); silviadattolo@hotmail.com (S.D.); noemimaria.giorgiano@studenti.unicampania.it (N.M.G.); andrea.demasi@studenti.unicampania.it (A.D.M.); francescoesposito-0101@hotmail.com (F.E.); mariaantonietta.puca@studenti.unicampania.it (M.A.P.); giuseppe.vicario@studenti.unicampania.it (G.V.); alfonso.fiorelli@unicampania.it (A.F.); 2Department of Mental Health and Public Medicine, Section of Infectious Diseases, University of Campania “Luigi Vanvitelli”, 80131 Naples, Italy; 3Department of Precision Medicine, University of Campania “Luigi Vanvitelli”, 80131 Naples, Italy; antonello.sica@fastwebnet.it

**Keywords:** surgery, nurse, thoracic surgery, nurses’ experiences in surgery, onco-hematological patients

## Abstract

**Background:** Nowadays, Thoracic Surgery is technologically advanced; therefore, it also focuses its attention on nursing care. The aim of the study is to evaluate the effect of the assessment of a dedicated team of nurses (DTN) in all onco-hematological patients undergoing VATS lobectomy for lung cancer on the outcome of the patient, preventing pressure injuries, reducing perioperative stress, duration of operations, complications, and hospital stay times. **Methods:** We performed a single-center observational retrospective study, including 31 DTN and 760 onco-hematological patients who underwent thoracic surgery between 30 October 2018 and 30 June 2023 at “Vanvitelli” University of Naples. **Results**: DTN ensures good nursing care before, during, and after surgery. Operative time was reduced by approximately 20 min, decreasing hospital infections in the DNT period and reducing intraoperative complications such as bleeding and hospital costs (*p* < 0.05). **Conclusions:** Thoracic surgery nurses require more specialized training to adapt to the development of sophisticated.

## 1. Introduction

Nowadays, lung cancer is the leading cause of cancer-related deaths worldwide. According to randomized clinical trials conducted in 1994, lobectomy is the standard surgical technique for lung-cancer patients in the early stages. This article focuses on the impact of the job of nurses in patients undergoing thoracic surgery, particularly in cases of a major lung resection [1].

Nurses have a fundamental role in every phase of preoperative, intraoperative, and postoperative thoracic surgery since the nurse’s task is to be the figure closest to the surgeon and the patient [2]. The requirement for highly qualified nurses has further increased in recent years related to technological and scientific progress and also improvement of competencies and expertise of nursing management, and the number of nurses both in the operating room and in other medical care services has also increased [3].

Therefore, optimizing the training process and ensuring the best professional nurses’ quality in the operating room in the shortest possible time has become fundamental and crucial in nursing education [4].

This article focuses on the role of nurses on patients in the operating room [1]. They have a fundamental role in the preoperative, intraoperative, and postoperative phases since the nurse is the figure closest to the surgeon and the patient [2]. Furthermore, the more complex and intensive characteristics of operating rooms, rigorous sterility, and strong specialization due to technological and scientific progress create a particularly high qualified demand for the workload of operating room nurses [3,5]. During surgery, anesthesiologists, surgeons, and nurses, all operating room professionals, have a direct impact on patient safety and health outcomes [6,7,8]. Therefore, optimizing the training process and ensuring the best professional quality of operating room nurses in the shortest possible time has become essential and crucial in nursing education [4,9].

Nurses are considered the main team and agent of change for the transformation of the health system to make it safer, and in the surgical environment, nurses play a key role in ensuring that best care practices provide patient safety [6,7,10,11]. We use the term dedicated nursing team (DNT) [12] to represent a nursing team that specializes in a subdiscipline of surgery, in our case thoracic surgery, and works consistently over a long period of at least nine months in a thoracic surgery operating room [8]. DNT are faced with multiple challenges daily, trying to meet the multiple needs of patients, their families, surgeons, and other members of multidisciplinary teams [13]. Operating room nurses have a fundamental role in creating optimal working conditions, ensuring a climate of safety and teamwork, reducing the level of stress, and clearly clarifying management’s perceptions [7,13], which could have an impact on the outcomes of the surgery. [3,8,9,10,11,14,15].

Therefore, the aim of the present study is to evaluate the effect of the assessment of a dedicated team of nurses (DTN) in all onco-hematological patients undergoing VATS lobectomy for lung cancer on the outcome of the patient, preventing pressure injuries, reducing perioperative stress, duration of operations, complications, and hospital stay times [11].

## 2. Materials and Methods

In our observational retrospective single center study, we evaluated the differences in work efficiency and onco-hematological patient outcomes who underwent thoracic surgery between a DNT period and a non-DNT period at the Thoracic Surgery Unit, Azienda Ospedaliera Universitaria (AOU) “L. Vanvitelli”, University of Campania L. Vanvitelli, Naples, from 30 October 2018 to 30 June 2023. All patients with onco-hematological disease who underwent thoracic surgery and for whom data regarding indicators of the quality of nursing work during the study period were available were included. We included all DTNs and non-DTNs that worked at the Thoracic Surgery Unit, and all consecutive patients undergoing thoracic surgery during the two periods were included. In detail, the inclusion criteria for patients were age > 18 years, respiratory and cardiac tests were within normal ranges, and no other preoperative comorbidities were indicated for surgery.

The exclusion criteria were severe neurological problems, recent myocardial infarction or unstable angina, emphysema, pregnancy, a prolonged prothrombin time (PT-INR) > 1.5 or a platelet count < 30,000, impossibility to tolerate single lung ventilation, and documented infection or severe sepsis diagnosed preoperatively in surgical patients; the exclusion criterion for nurses was maternity leave or breastfeeding.

The study was conducted according to the guidelines of the Declaration of Helsinki 1975, revised in 1983, and the rules of the Italian laws of privacy and approved by the Institutional Ethics Committee of “Universita” Della Campania “Luigi Vanvitelli”-AOU “L. Vanvitelli” (protocol number 766/2018, 7 March 2018). Informed consent was obtained from all subjects involved in the study, in detail written informed consent has been obtained from the patient(s) for clinical investigations and scientific purposes to publish this paper.

The DTN was organized by professional team leaders, and team members were professional nurses working as a thoracic surgery team for a period of at least nine months. Surgical procedures in thoracic surgery are specialized, complex, and precise [11], and it is difficult for nurses to fully master the knowledge related to the procedure in a short period. Thoracic surgery nurses must recognize the complexity of their work and design all preparation phases to ensure that the DTN can meet the challenges of complex and delicate work. We used a “standardized training” model, which provides hierarchical professional training in various stages of preparation.

The climate of safety and comfort was assessed using the surgeon satisfaction survey designed by the department. A questionnaire was administered at the end of each intervention to all the staff in the room to assess the peculiarities of the nurses, sterility, post-operative surgical equipment, and the incidence of accidental needlestick sticks among the medical staff, the electrical burns among the patients, and the incidence of patients falling or falling out of bed in all 760 patients, divided into two groups depending on whether they underwent nursing management between non-DNT and DNT. By surgeon’s satisfaction we mean the reduction in stress in the operating room: empathy with the nursing staff who know all the operating times and the devices that are used in the room during surgery; therefore, adequate sutures, adequate staplers, careful monitoring of saturation, intraoperative blood loss, diuresis, and heart rate.

In the DTN and non-DTN groups, nurses with standardized training were regularly rotated through different specialty clinics every nine months [16].

During the intraoperative phase, the nurses first invite the patient to go to the operating room, offering him all the psychological and emotional support he needs; carry out blood chemistry tests; they administer antibiotic therapy as prophylaxis before the incision. They prepare the skin: first with a shower with simple soap, then with hair removal of the armpits and chest, immediately before the operation, and finally with the use of a skin antiseptic solution with a chlorhexidine-alcohol solution [17,18,19].

Nurses provide assistance to anesthesiologists for thoracic epidural analgesia and multimodal analgesic strategies, performed according to anesthetic guidelines [20].

They provide warming to the patient to prevent intraoperative hypothermia. Hypothermia has been shown to alter drug metabolism, adversely affect coagulation, increase the risk of bleeding, and increase cardiac morbidity and wound infection.

Post-operative chills also increase oxygen consumption and worsen pain [21]. Nurses take care of the preparation of all the instruments and devices necessary to follow the lobectomy in video-assisted thoracoscopic surgery (VATS), organizing and planning all the operating times [22].

They take care of the patient’s venous and bladder catheterization, assist the anesthetists during selective intubation, help the anesthetists during all lung exclusion maneuvers, assist the intubation with the aid of the flexible bronchoscope (Figure 1), ensure the correct positioning of the patient on the bed operating, in lateral decubitus, the correct positioning of legs and arms, with a correct inclination of the operating bed (Figure 2 and Figure 3) guarantee careful and constant monitoring of vital parameters throughout the surgical procedure, providing an accurate relationship between saturation, heart rate, blood pressure, diuresis, blood loss, and air loss [23].

They provide adequate suture threads and staplers, correctly differentiate parenchymal, vascular, and bronchial staplers. They assist the surgeon during all phases of the operation [24]. They take care of the patient’s fluid balance by administering balanced intravenous fluids during surgery: intraoperative crystalloids 550–1050 mL or crystalloids, 1550 mL, collide 530–1060 mL. They deal with the prophylaxis of postoperative nausea and vomiting only in patients defined as high risk based on preoperative screening [25]. Nursing staff play a critical role in implementing enhanced recovery protocols; the success of the execution of this new path is certainly linked to the close collaboration with other healthcare professionals [26].

### 2.1. Surgical Techniques

Video-assisted thoracoscopic surgery is the cornerstone of minimally invasive thoracic surgical approaches and is the gold standard in the surgical treatment of lung cancer.

All patients underwent general anesthesia and single-lung ventilation using a double-lumen endotracheal tube so that the lung collapsed completely; therefore, a thoracoscope was inserted into the pleural space.

The three-portal approach, according to Hansen et al. [27], was used. It consists of a 4–5 cm port anterior incision, in the 4th intercostal space, between the breast and the lower corner of the scapula anterior to the latissimus muscle, along the anterior axillary line for a utility incision and two 1–1.5 cm inferior access incisions, located in the 7th or 8th intercostal space, respectively, along the anterior to the hilum, at the level of the diaphragm, and posterior axillary lines in a straight line starting from the shoulder blade, for two thoracoscopic ports. The camera is inserted into the lower door angled at 30° (Figure 4).

The hilar structures must be dissected individually, a standard lymphadenectomy is necessary, and the surgical piece must be removed with an endobag.

Lobectomy can be performed with a “hilum first” or “fissure first” approach, depending on the surgeon’s experience; in addition, the hilar structures are individually dissected and a standard lymphadenectomy, as in open surgery, is mandatory.

This minimally invasive approach is safe, feasible, and oncological efficient, reducing complications, a better quality of life after surgery, ensuring pain control, better esthetic results offering faster recovery and finally a better cost-effectiveness ratio (Figure 4).

### 2.2. Statistical Analysis

The summary statistics of patients’ characteristics were tabulated either as mean ± standard deviation (SD) for continuous variables or as the number of patients and percentages for categorical variables.

Patients were assessed in two periods depending on whether they underwent nursing management between the non-DNT period and the DNT period. We use the term dedicated nursing team (DNT) to represent a nursing team that specializes in thoracic surgery and works consistently over a long period of at least nine months in this setting.

Student’s *t*-test and chi-squared test were used to compare different variables as appropriate. Confidence intervals (CIs) at 95% and 2-sided *p*-values were calculated.

The MedCalc statistical software (Version 12.3) (MedCalc Software, Ostend, Belgium) was used. The *p*-value statistically significant was considered if it was low of 0.05.

## 3. Results

We performed a single-center observational retrospective study, including 65 nurses, all of whom worked at our Thoracic Surgery Unit: 31 Nurses in DNT and 34 in non-DNT group (Table 1).

The mean age of the nurses in DNT was 32.1 years old; 20 (64.5%) nurses were between 32 and 36 years old. The prevalence of female sex in 28 (91.5%) nurses, 29 (93.5%) nurses had a bachelor’s degree, of whom 27 (93.1%) had a master’s degree in operating room staff (Table 1).

The average years of total employment were 7.75 ± 3.5 in the Thoracic Surgery Unit. The prevalence of nurses (67.7%) had acquired good work experience in the Thoracic Surgery Unit; in detail, 7 (33.3%) nurses had been working for 6–10 years and 14 (66.7%) nurses for 11–15 years at the same Thoracic Surgery Unit (Table 1).

The mean age of the nurses in non-DNT was 43.3 years old, 15 (44.1%) nurses were between 40 and 49 years old, and more than 50 years old in 17 (50%) nurses. Twenty-one (64.5%) nurses were female; 9 (26.5%) nurses had a bachelor’s degree, of whom 2 (22.3%) had a master’s degree in operating room staff after a bachelor’s degree (Table 1). The average years of total employment were 4.5 ± 3.2 in the Thoracic Surgery Unit (Table 1).

The prevalence of nurses who acquired good work experience in the Thoracic Surgery Unit was 11.8%; in detail, 4 (100%) nurses had been working for 6–10 years.

In Table 1 are reported the statistical differences between the 31 nurses in DNT and the 34 nurses in the non-DNT group: nurses in DNT are younger, with a higher prevalence of nurses with a bachelor’s degree (*p* < 0.05) and a master’s degree in operating room staff (*p* < 0.05), with more average years of total employment in the Thoracic Surgery Unit *p* < 0.05), and with a better acquired good work experience in the Thoracic Surgery Unit (*p* < 0.05) (Table 1).

In Table 2 are reported characteristics of all 760 onco-hematological patients who underwent thoracic surgery enrolled in the study.

The prevalent comorbidity observed was Hypertension arteriosus in 88.2% (625) of cases, followed by diabetes mellite in 73.5%, Chronic Obstructive Pulmonary Disease in 59.6% cardiac disorders in 37.6%, and asthma in 15.8% (Table 2).

All the 760 patients were assessed in two groups depending on whether they underwent nursing management between non-DNT and DNT: 295 patients with no-DNT and 465 patients with DNT (Table 3).

In Table 3 are reported the characteristics of all 760 observed patients divided into two groups depending on whether they underwent nursing management between the non-DNT and DNT group: patients observed with non-DNT and patients observed with DNT.

There is a significant difference between patients observed with DNT and non-DNT group in relation to nurses’ peculiarities, including surveillance rates on surgical patients, location of surgery, previous allergies, expected surgical time, antibiotic use 60 min before incision, sterility, availability of surgical equipment and surgical materials, inventory of surgical instruments, type of surgery, surgical specimen, postoperative surgical equipment, remaining surgical foreign bodies, presence of pressure ulcers during surgery, use of perioperative medications, unplanned extubating, transfusion reaction during the surgical period, incidence of needle sticks among medical personnel, incidence of electrical burns among surgical patients, and incidence of surgical patients falling or falling out of bed.

We observed a prevalence of male both in patients observed with non-DNT and in those with non-DNT (221 (74.9%) males versus 74 (25.1%) female patients and 383 (82.4%) males versus 81 (17.6%) female patients, respectively, *p* < 0.05) (Table 3).

The incidence of hospital infections is lower in the DNT group compared to non-DNT (6% versus 25%, *p* < 0.05) (Table 3).

According to the etiology of infection, a total of 40 (5.3%) Gram-positive bacilli, 46 (6.0%) Gram-negative bacilli, and 14 (1.8%) fungi were identified, and we found that the rate of Gram-positive and Gram-negative bacilli and fungal infection was lower in patients with DNT period than in the preceding non-DNT period (*p* < 0.05) (Table 3).

The turnover rate (dichotomous variable: 1 = more than 30 min; 0 = less than 30 min) was more efficient in DNT than in non-DNT (*p* ≤ 0.05) (Table 3).

The times of the entire surgical procedure (including anesthetic times) were prolonged in non-DNT patients than in patients observed with DNT (*p* < 0.05).

We reported a statistically significant reduction in operation time in DNT compared to the non-DNT group (101.2 ± 23.5 versus 150.4 ± 20.5 min, *p* < 0.05).

A statistically significant reduction in the number of days of hospitalization (*p* << 0.05), post-operative days of hospitalization (*p* < 0.001), time spent in the thoracic drainage tube (*p* < 0.05), and hospital costs of the surgical procedure (*p* < 0.05) were observed in patients with non-DNT.

The intraoperative complications were more prevalently reported in patients observed with no-DNT period than in those observed with DNT with a specialized DNT nursing team (55 (18.6%) versus 28 (6.02%), respectively, *p* < 0.05).

In the group of patients observed with DNT, an improvement in the safety and comfort climate were reported (*p* < 0.05), in detail for the level of surgeon satisfaction, for localization surgery, and for the expected surgical time, while in the group of patients observed in the non-DNT period, only an improvement in surgeon satisfaction and the expected surgical time was observed (Table 4).

An improvement in sterility in terms of availability of surgical equipment and material was observed only in the group of patients observed with DNT (*p* <0.05), as well as the inventory of surgical instruments for the specific type of surgery, i.e., the correct supply of the material (Table 4).

On the other hand, an increased incidence of accidental needlestick punctures among medical personnel, electric burns among patients, and an increase in the incidence of patients falling or falling out of bed was observed only in the group of patients observed in the non-DNT group (Table 4).

## 4. Discussion

DNTs better perform WHO-recommended standardized surgical procedures specific to the intraoperative and postoperative phases because each team member knows exactly and precisely all the phases of the surgery and can think better about the future and prepare for possible complications in a timely manner [28,29,30]. Furthermore, DNTs have being able to deal with complications such as vascular injury, being able to provide the surgeon with stitches and a stapler quickly, helping the surgeon to resolve bleeding; reduce intraoperative infections by ensuring greater sterility, and correct one-lung ventilation while also constantly assisting the anesthetist.

DTN guarantees the internal working serenity of all staff, since surgeon satisfaction is linked to productivity and patient results [31,32]. Effective surgeries and safe and precise operations in thoracic surgery require highly professional teamwork [33], and the stability provided by a specialized team can improve teamwork performance [34].

However, there is a risk of complications caused by the tension between the conflicting goals of efficiency and safety in thoracic surgery [35]. Nurses in operating rooms face strenuous and complex workloads daily, but more importantly, they face many professional risks, which can quickly lead to mental and physical exhaustion. In recent research, operating room nurses have been shown to suffer from more work-related stress and higher levels of depression and anxiety than ward nurses [36]. The use of a dedicated team should improve efficiency, patient safety, and management of critical events [37,38].

However, surgical training and courses are not part of the regular curriculum of most nursing schools [39], and it may be difficult for surgeons to tolerate the difficulties of an inexperienced nurse during surgery because surgeons and nurses rarely have training joint. as a surgical team [40]. Furthermore, there is currently no standardized training, which prevents new nurses from understanding the role of thoracic surgery nurses during surgical procedures [41,42].

According to reflective life, training a team of specialists is recommended to prevent surgical site infections (SSI) and ensure patient safety during intraoperative care [43]. Furthermore, it has been reported that the peculiarity of the nursing team contributes to the reduction in operating times, suggesting potential benefits in maintaining continuity of membership of the team over time [44].

Thoracic surgery nurses provide professional nursing care in surgery and can play an important role in reducing surgery time and improving the quality of surgery and patient outcomes [44,45,46,47]. The importance and value of nurses’ work are directly reflected in nursing quality; therefore, specific indicators are essential to evaluate the quality of their work and can provide valuable insights to improve their clinical practice [48,49].

Nurses working in the operating room have more knowledge than nurses working in any other department. This may be due to the quality of in-service training received or the greater number of nurses holding a bachelor’s degree [50].

A statistically significant difference was observed between the age groups of nurses. The higher level of knowledge of the younger nurses in the present study could be associated with the young nurses having not yet forgotten the knowledge acquired during their studies. It should also be considered that some had also acquired a master’s degree; this highlights the desire to improve continues. Nurses who had completed undergraduate or postgraduate programs were found to have greater knowledge than other nurses. Also, Kallman et al. [51] and Lawrence et al. [52] demonstrated similar results. Variability in knowledge in the nursing setting is influenced by age, years of experience, and ability to access up-to-date sources of evidence; 54% of nurses had searched for medical information via the Internet, and these had a higher level of knowledge than others [42].

In addition, the enthusiasm of the first years of work is greater, especially in young people and in stimulating areas such as surgery. A close-knit team is a winning weapon in every field. This may be due to the increased length and quality of performance improvement training that accompanies a higher level of education and improves the quality of healthcare. We observed that the performance and total knowledge of nurses who had worked for 6–10 years were higher than those of nurses who had worked for 11–15 years, or for 21 years and more. However, Lawrence et al. [52] stated that knowledge scores increased with increasing years of experience.

Iranmanesh et al. [53] failed to find a relationship between years of experience and knowledge scores. The higher scores for knowledge of PI risk factors among younger nurses may be due to the emphasis on ORPI risk prevention in subsequent formal education and the lack of continuous learning.

In our study, professional nurses had relatively long work experience (average 15 years), which may have influenced the results. It has been argued that as professionals gain more experience, commonly performed deliberate tasks become more automatic, which can make complex work easier [54,55].

Furthermore, since the operating room department of our study served as a teaching hospital, student nurses were present during the thoracic surgery sessions, explaining their tasks and duties and supervising their work. Providing orientation to younger people is certainly essential for training dedicated nurses in complex surgical interventions such as thoracic surgery. In our study, the less experienced nurses are guided by the more experienced nurses; their knowledge allows the surgeon to work with more serenity, reducing tension and stress since he correctly knows all the phases of the surgical intervention [56]. Their experience has demonstrated in our study the reduction in operating costs because they know exactly the materials and devices (staplers, sutures, etc.) that are needed and the reduction in operating and anesthetic times [57].

Griffiths et al. [58] in a systematic review of 23 observational studies noted that increasing the number of skilled nurses can be cost-effective. In fact, in acute care hospitals, high levels of nursing skills correlate with improved quality of care and a reduced risk of death. On the other hand, low-consciousness nursing staff in general medicine and surgery units increase the risk of patient death by 2–3% [59].

However, this study is not without limitations. First, it is not a randomized study. Secondly, we could not evaluate the procedure duration of each surgery using reliable measures; this is a single-center study. Finally, the fixed team of nurses has not been fully separated from non-DNT nurses due to variations in the number of surgeries performed in each specialty each day, and some nurses may not be well adapted to the ongoing development of cross-specialty surgical procedures [60,61,62,63,64,65,66].

## 5. Conclusions

The development of leadership of specialist nurses in thoracic surgery guarantees greater efficiency of the nursing team and a good climate of safety, reliability, efficiency, and serenity in the surgical team, obtaining better results for patients, reducing turnover times, and improving surgeon satisfaction, correlated with improved productivity.

Thoracic surgery nurses require more specialized training and practice to adapt to the development of increasingly sophisticated surgical techniques.

The benefit of working in a long-term stable team includes increased trust between nurses and surgeons, which translates into better performance; therefore, surgical teams should involve all members of the surgical ecosystem, and the safety environment is better in the context of good teamwork.

Therefore, with a specialized DNT nursing team, intraoperative complications are reduced compared to the no-DTN team because expert nurses actively participate alongside the surgeon, reducing the risk of bleeding, because they intervene promptly by providing the surgeon with adequate devices and hemostatics for control of bleeding. In conclusion, the old concept of “observe and learn” must be replaced with a new concept of “observe and do to learn”. This represents a milestone for young nurses in training for a full and complete professional experience.

## Figures and Tables

**Figure 1 healthcare-12-01843-f001:**
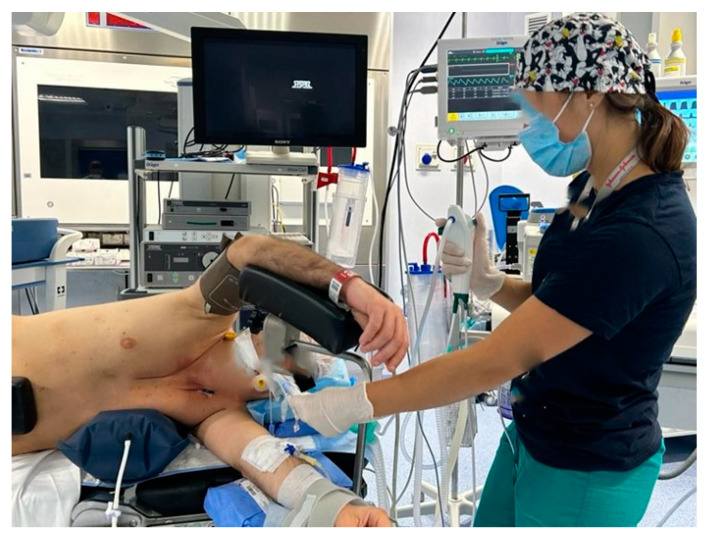
Nurses assist the anesthetists during selective intubation, help the anesthetists during all lung exclusion maneuvers.

**Figure 2 healthcare-12-01843-f002:**
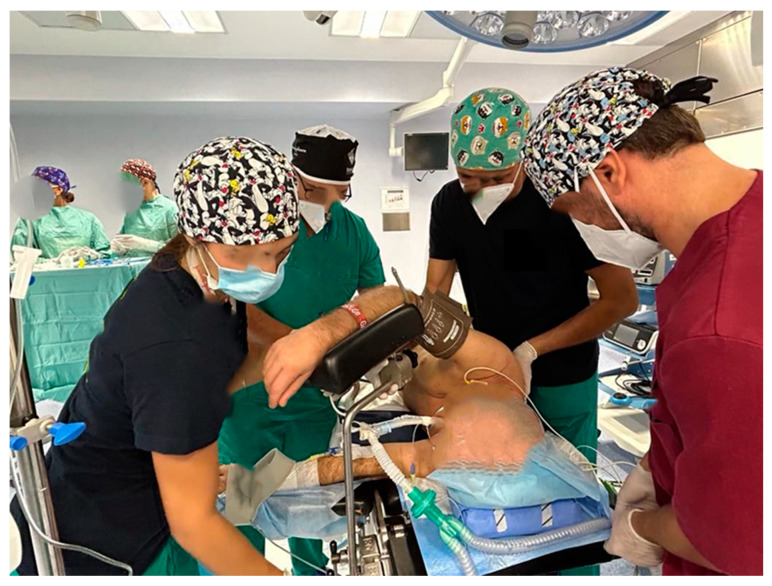
Nurses ensure the correct positioning of the patient on the bed operating, in lateral decubitus.

**Figure 3 healthcare-12-01843-f003:**
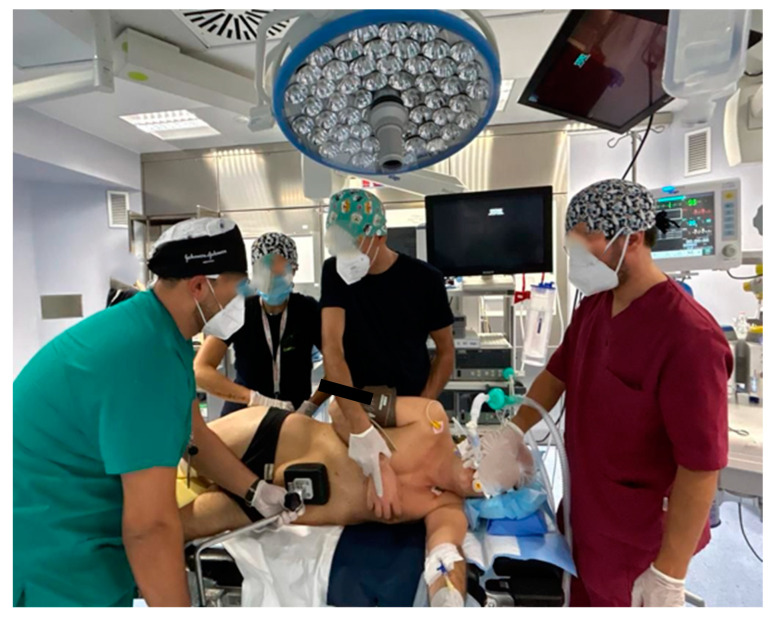
The correct positioning of legs and arms with a correct inclination of the operating bed.

**Figure 4 healthcare-12-01843-f004:**
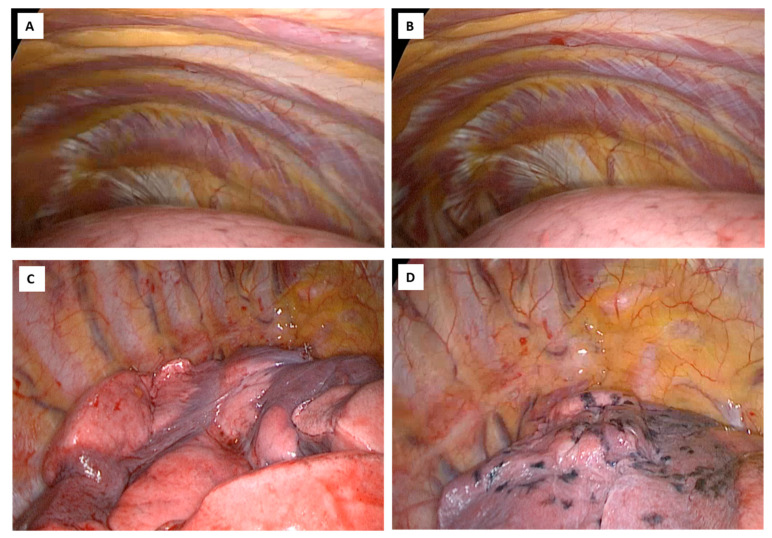
(**A**): The three-portal approach used according to Hansen et al. [27]. (**B**): A 4–5 cm port anterior incision in the 4th intercostal space for a utility incision. Two 1–1.5 cm inferior access incisions, located in the 7th or 8th intercostal space, respectively, along the anterior and posterior axillary lines for two thoracoscopic ports. (**C**,**D**): All patients underwent general anesthesia and single-lung ventilation using a double-lumen endotracheal tube so that the lung collapsed completely.

**Table 1 healthcare-12-01843-t001:** Characteristics of all 31 nurses in DNT and 34 in non-DNT group.

	Nurses in DNT	Nurses in Non-DNT	*p*-Value
Number of Nurses	31	34	
Female, n (%):	28 (91.5)	21 (61.8)	<0.05
Age, years, mean ± SD	32.1 ± 15.1	43.3 ± 3.9	<0.05
Age, 32–36 years old, n (%):	20 (64.5)	2 (5.9)	<0.05
Age, 40–49 years old, n (%):	8 (25.8)	15 (44.1%)	
Age, >50 years old, n (%):	3 (9.7)	17 (50%)	
Nurses with a bachelor’s degree, n (%):	29 (93.5)	9 (26.5)	<0.05
Nurses with a master’s degree in operating room staff after bachelor’s degree, n (%):	27 (93.1)	2 (22.3)	<0.05
The average years of total employment in the Thoracic Surgery Unit, mean ± SD	7.75 ± 3.5	4.5 ± 3.2	<0.05
Nurses acquired good work experience in the Thoracic Surgery Unit, n (%):	21 (67.7)	4 (11.8)	<0.05
nurses working for 6–10 years	7 (33.3)	4 (100)	
nurses working or 11–15 years	14 (66.7)	0	

**Table 2 healthcare-12-01843-t002:** Characteristics of all 760 onco-hematological patients who underwent thoracic surgery, between 30 October 2018 and 30 June 2023.

Number of subjects	760
Sex, n (%):	
Male	604 (79.5)
Female	156 (20.5)
Age, years, mean ± SD (range)	65.2 ± 3.2 (52–76)
Smokers (yes), n (%)	453 (59.6)
BMI, kg/m^2^, mean ±SD:	24.5 ± 2.3
Comorbidities, n (%):	
Hypertension	625 (88.2)
Diabetes	557 (73.3)
Cardiac disorders	285 (37.6)
Chronic Obstructive Pulmonary Disease	430 (59.6)
Asthma	120 (15.8)
Localization of the lung nodule, n (%):	
left upper lobe (lul)	162 (21.3)
Left lower lobe (lll)	214 (28.2)
Rigt upper lobe (rul)	139 (18.3)
Right lower lobe (rll)	132 (17.4)
Middle lobe (mL)	113 (14.8)
Surgical resections, n (%):	
Lul	141 (18.5)
Lll	251 (33.0)
L s4 + s5	6 (0.8)
L s6	3 (0.4)
rul	186 (24.3)
rll	161 (21.8)
Bilobectomy	9 (1.2)
Wedge resection	3 (0.4)
Histological Characteristics, n (%):	
Adenocarcinoma	365 (48.0)
Squamous Carcinoma	260 (34.2)
Adenosquamous Carcinoma	44 (5.8)
Pleomorphic Carcinoma	9 (1.2)
Typical Carcinoids	3 (0.4)
Atypical Carcinoids	7 (0.9)
Primary Oncological Disease, n (%):	
B-Cell Lymphomas	36 (4.7)
Inflammatory Myofibroblastic Tumor	15 (1.97)
Secondary Oncological Disease, n (%):	
Colorectal Metastases	21 (2.7)
Pre-operative laboratory data, mean ± SD:	
Albumin (g/dL)	3.5 ± 0.7
Protein (g/dL)	6.5 ± 2.3
White blood cells (/uL)	7387.3 ± 3352.9
Hemoglobin (g/dL)	12.3 ± 3.1

**Table 3 healthcare-12-01843-t003:** Characteristics of all 760 patients divided into two groups depending on whether they underwent nursing management between the non-DNT and the DNT.

	Patients Observed with Non-DNT	Patients Observed with DNT	*p*-Value
Number of subjects	295	465	
Sex, n (%):			<0.05
Male	221 (74.9)	383 (82.4)	
Female	74 (25.1)	82 (17.6)	
The incidence of hospital infections, n (%):	74 (25)	26 (6)	<0.05
GRAM-positive bacilli infection, n (%):	28 (37.8)	12 (46.1)	<0.05
GRAM-negative bacilli infection, n (%):	36 (48.6)	10 (38.5)	<0.05
Fungal bacilli infection, n (%):	10 (13.5)	4 (15.3)	<0.05
The turnover rate			
(dichotomous variable: 1 = more than 30 min; 0 = less than 30 min)	1	0	<0.05
The times of the entire surgical procedure (including anesthetic times), minutes, mean ± SD,	39.7 ± 8.6	20.2 ± 4.8	<0.05
Operation time (minutes), mean ± SD,	150.4 ± 20.5	101.2 ± 23.5	<0.05
Days of hospitalization (days), mean ± SD,	6.5 ± 9.6	4.3 ± 3.5	<0.05
Post-operative stay (days), mean ± SD	4.5 ± 6.3	2.2 ± 3.6	<0.05
Chest tube duration time (days), mean ± SD	6.2 ± 3.2	2.2 ± 2.4	<0.05
Hospital costs of surgery, EURO, mean ± SD	8700.0 ± 1500.00	7200.0 ± 850.0	<0.05
Blood loss, n (%)	5 (1.7)	7 (1.5)	
Blood loss, mL, mean ± SD	255.6 ± 54.3	231.2 ± 79.6	
Chest drainage, mL daily, mean ± SD	103.8 ± 78.2	110.2 ± 67.2	0.1886
Chest tube duration, days, mean ± SD	5.4 ± 2.8	3.2 ± 1.3	<0.05
Need for blood transfusion, n (%)	0	0	--
Complications, n (%):			
Air leaks	22 (7.5)	15 (3.3)	<0.05
Atelectasis	33 (11.2)	13 (2.8)	<0.05
Nurses’ peculiarities expertise	0	1	<0.05

**Table 4 healthcare-12-01843-t004:** Evaluation of safety and comfort climate, nurses’ peculiarities, sterility, post-operative surgical equipment, and the incidence of accidental needlestick punctures among medical personnel, electric burns among patients, and the incidence of patients falling or falling out of bed in all the 760 patients divided into two groups depending on whether they underwent nursing management between non-DNT and DNT.

	Patients Observed with Non-DNT	Patients Observed with DNT	*p*-Value
Number of subjects	295	465	
	Improvement yes	Improvement yes	
Improvement of climate of safety and comfort designed by department, (1 = improvement yes; 0 = no):			
Surgeon satisfaction surgery	41%	78%	<0.05
Location surgery	45%	75%	<0.05
Expected surgical time	30%	72%	<0.05
Improvement of Nurses’ peculiarities, (1 = improvement yes; 0 = no):			
Surveillance rates on surgical patients (previous allergies, antibiotic use 60 min before incision)	51%	87%	<0.05
Improvement of Sterility, (1 = improvement yes; 0 = no):			
Availability of equipment and surgical materials	43%	79%	<0.05
Inventory of surgical instruments, (1 = yes; 0 = no):			
Type of surgery	2%	10%	<0.05
Surgical specimen	31%	51%	<0.05
Post-operative surgical equipment, (1 = yes; 0 = no):			
Surgical foreign bodies	None	None	--
Pressure ulcers during surgery	None	None	--
Use of perioperative medications	100%	0	ns
Unplanned extubation	None	None	--
Transfusion reaction	None	None	--
Incidence (1 = yes; 0 = no):			
Needle sticks among medical personnel	39%	3%	<0.05
Electrical burns among surgical patients	25%	2%	<0.05
Surgical patients falling out of bed	4%	0	<0.05

ns: not significant.

## Data Availability

Data are contained within the article.

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
