# Peer review of "The Impact of Assessment of Nurses’ Experiences in Thoracic Surgery in Onco-Hematological Patients"

_healthcare, 2024, doi:10.3390/healthcare12181843_

Round 1

Reviewer 1 Report (Previous Reviewer 2)

Comments and Suggestions for Authors

Thank you very much for your article and the opportunity to evaluate it. Below, I present the scientific and methodological criteria to improve your article.

In summary, the abstract of the article requires significant revision to enhance the clarity of the objectives, provide a more detailed description of the methodology used, present the results coherently, and adequately support the conclusions with the study findings.

The introduction has improved regarding the importance of the topic and has been contextualized, also justifying how nurses' experiences affect patient care.

The objective is clearer, although it could be more specific regarding how each outcome is measured.

Methodology and results are clear, providing specific effect measures and confidence intervals, and corrected p-values. Check that all digits in p-values are the same, whether you enter 2 or 3.

Areas for improvement include more detailed information on study design, participant selection criteria, data collection procedures, and analysis.

Clarity of the results has been improved; I agree that the images are not suitable for a scientific article.

Future directions need further refinement.

Overall, it is a good study.

Author Response

To the Editor in Chief of Healthcare

We re-submit our article “The Impact Of Assessment Of Nurses’ Experiences In Thoracic Surgery In Onco-Hematological Patients ”, Manuscript ID: healthcare-3115321

The following changes (shown underlined). The manuscript has been improved according to the suggestions of the reviewer: I thank the Editor for the opportunity of reviewing this paper.

Reviewer(s)' Comments to Author:

Reviewer #1: Thank you very much for your article and the opportunity to evaluate it. Below, I present the scientific and methodological criteria to improve your article.

POINT 1:  In summary, the abstract of the article requires significant revision to enhance the clarity of the objectives, provide a more detailed description of the methodology used, present the results coherently, and adequately support the conclusions with the study findings.

Answer to the Reviewer point 1: The observation of the reviewer has been accepted and the new manuscript has been  modified accordingly. (lines:24,25,27-31)

POINT 2: The introduction has improved regarding the importance of the topic and has been contextualized, also justifying how nurses' experiences affect patient care.
Answer to the Reviewer point 2: We thank the Reviewer for observation.(lines 72-81)

Point 3: The objective is clearer, although it could be more specific regarding how each outcome is measured.

Answer to the Reviewer point 3: The observation of the reviewer has been accepted and the new manuscript has been  modified accordingly. lines: 71-82

POINT 4: Methodology and results are clear, providing specific effect measures and confidence intervals, and corrected p-values. Check that all digits in p-values are the same, whether you enter 2 or 3.
Answer to the Reviewer point 4: The observation of the reviewer has been accepted and the new manuscript has been  modified accordingly. (table 1,3,4 and lines:32,236,279, 281, 283,284,286)

POINT 5: Areas for improvement include more detailed information on study design, participant selection criteria, data collection procedures, and analysis.
Answer to the Reviewer point 5: We thank the Reviewer for observation.

POINT 6: Clarity of the results has been improved; I agree that the images are not suitable for a scientific article.

Future directions need further refinement.
Overall, it is a good study

Answer to the Reviewer point 6: We thank the Reviewer for observation. The observation of reviewer has been accepted. We would like to point out that images can help to understand the importance of the role of a Nurses assist specialized in the phases of patient management in the operating room.

Reviewer 2 Report (Previous Reviewer 4)

Comments and Suggestions for Authors

Dear authors,

Thank you for resubmitting the revised manuscript and addressing all of my comments.

Author Response

To the Editor in Chief of Healthcare

We re-submit our article “The Impact Of Assessment Of Nurses’ Experiences In Thoracic Surgery In Onco-Hematological Patients ”, Manuscript ID: healthcare-3115321

The following changes (shown underlined). The manuscript has been improved according to the suggestions of the reviewer: I thank the Editor for the opportunity of reviewing this paper.

Reviewer(s)' Comments to Author:

Reviewer #2: Dear authors,
Thank you for resubmitting the revised manuscript and addressing all of my comments.

Answer to the Reviewer 2: We thank Reviewer for helping us to improve our paper.

Reviewer 3 Report (Previous Reviewer 5)

Comments and Suggestions for Authors

thanks for your revision

and attention to details

great work

Author Response

To the Editor in Chief of Healthcare

We re-submit our article “The Impact Of Assessment Of Nurses’ Experiences In Thoracic Surgery In Onco-Hematological Patients ”, Manuscript ID: healthcare-3115321

The following changes (shown underlined). The manuscript has been improved according to the suggestions of the reviewer: I thank the Editor for the opportunity of reviewing this paper.

Reviewer(s)' Comments to Author:

Reviewer #3: thanks for your revision, and attention to details great work

Answer to the Reviewer 3 : We thank Reviewer for helping us to improve our paper.

Reviewer 4 Report (New Reviewer)

Comments and Suggestions for Authors

Dear authors:

I reviewed your paper entitled “The Impact Of Assessment Of Nurses’ Experiences In Thoracic

Surgery In Onco-Hematological Patients”.  The study aims: to demonstrate the impact of a dedicated team of nurses (DTN) on outcome of patient, preventing pressure injuries, reducing perioperative stress, duration of operations, complications and hospital stay times.

 First, I would like to congrats the authors, it is very interesting thematic.

In order to improve your paper, I have some suggestions:

·        -  In abstract:

v  please reflect about the aim, you insert in method, and I suggest you reflect about the way to present your aim: could be: To evaluated effect of assessment of DNT in all onco-hematological patients undergoing VATS lobectomy for lung cancer, instead of “To demonstrate the impact of a dedicated team”;

v  Insert in method the type of study not in results.

v  Please reflect about the term “impact” you are not doing an impact study, you done a single-center observatinal retrospective study.

v  In method you should mention the two groups in the study (DNT en non-DNT); Only use the  inclusion/exclusion criteria in the paper not in abstract.

·        - In page 2: the aim of the study should be described as in abstract.

·        - In 2.2. point you describe the statistic tests used, what were the main decisions to use parametric and non-parametric tests?

·         -Line 239 (results) you have information that should be only in methods (e.g. time of study developing) not in results.

I have nothing to add, and I wish you good luck towards publishing it!

Best regards.

Author Response

To the Editor in Chief of Healthcare

We re-submit our article “The Impact Of Assessment Of Nurses’ Experiences In Thoracic Surgery In Onco-Hematological Patients ”, Manuscript ID: healthcare-3115321

The following changes (shown underlined). The manuscript has been improved according to the suggestions of the reviewer: I thank the Editor for the opportunity of reviewing this paper.

Reviewer(s)' Comments to Author:

Reviewer #4: Dear authors:
I reviewed your paper entitled “The Impact Of Assessment Of Nurses’ Experiences In Thoracic
Surgery In Onco-Hematological Patients”.  The study aims: to demonstrate the impact of a dedicated team of nurses (DTN) on outcome of patient, preventing pressure injuries, reducing perioperative stress, duration of operations, complications and hospital stay times.
First, I would like to congrats the authors, it is very interesting thematic.
In order to improve your paper, I have some suggestions:
·        POINT 1:  In abstract:
v  please reflect about the aim, you insert in method, and I suggest you reflect about the way to present your aim: could be: To evaluated effect of assessment of DNT in all onco-hematological patients undergoing VATS lobectomy for lung cancer, instead of “To demonstrate the impact of a dedicated team”;
v  Insert in method the type of study not in results.
v  Please reflect about the term “impact” you are not doing an impact study, you done a single-center observatinal retrospective study.
v  In method you should mention the two groups in the study (DNT en non-DNT); Only use the inclusion/exclusion criteria in the paper not in abstract.

Answer to the Reviewer point 1: The observation of the reviewer has been accepted and the new manuscript has been  modified accordingly.

POINT 2:   In page 2: the aim of the study should be described as in abstract.
·        - In 2.2. point you describe the statistic tests used, what were the main decisions to use parametric and non-parametric tests?Answer to the Reviewer point 2: The observation of the reviewer has been accepted and the new manuscript has been  modified accordingly.
In detail to Evaluate of safety and comfort climate, nurses’ peculiarities, sterility, post-operative surgical equipment, and the incidence of accidental needlestick punctures among medical personnel, electric burns among patients, and the incidence of patients falling or falling out of bed in all the 760 patients divided into two groups depending on whether they underwent nursing management between non-DNT and DNT we used non-parametric tests using a questionare

POINT 3:  Line 239 (results) you have information that should be only in methods (e.g. time of study developing) not in results.
I have nothing to add, and I wish you good luck towards publishing it!
Best regards.

Answer to the Reviewer point 3: The observation of the reviewer has been accepted and the new manuscript has been  modified accordingly.

We thank the Editor and the Reviewers for helping us to improve our paper.
The manuscript has been read and approved by all the authors and has not been submitted for publication to other journals. We also declare that we have no conflict of interest in connection with this paper. Neither the manuscript nor part of it has been published or is under consideration for publication elsewhere. We also declare that we have no conflict of interest in connection with this paper.
The address for correspondence is: Prof.ssa Caterina Sagnelli, Department of Mental Health and Public Medicine, University of Campania “Luigi Vanvitelli”, 80131 Naples, Italy, Tel: +393938107860, fax: +390815666477, 80131, e-mail: caterina.sagnelli@unicampania.it; sagnelli.caterina@libero.it
Please use the voucher required for Dr. Antonello Sica as Editor Healthcare.

We sincerely hope that the enclosed manuscript can be accepted for publication in the: Healthcare

Prof.ssa Caterina Sagnelli

Dr. Antonello Sica

This manuscript is a resubmission of an earlier submission. The following is a list of the peer review reports and author responses from that submission.

Round 1

Reviewer 1 Report

Comments and Suggestions for Authors

The paper has very low scientific soundness and I think it has low interest to the readers. The novelty of the paper is low.

Comments on the Quality of English Language

The paper require editing of English language 

Reviewer 2 Report

Comments and Suggestions for Authors

Thank you very much for your article and the opportunity to evaluate it. Below, I present the scientific and methodological criteria to improve your article.

The abstract presents several errors and inconsistencies:

Unclear objective: Although it mentions the objective of demonstrating how a dedicated team of nurses can influence various aspects of thoracic surgery, such as the work of surgeons and anesthesiologists, perioperative stress, postoperative complications, etc., the abstract fails to provide a clear and specific statement of the research objective. It should be more precise and clearly define what is intended to be demonstrated or investigated.

Insufficient methodology: Although it states that a single-center retrospective study was conducted, it does not provide detailed information on how the study was carried out. There is no mention of how the data were collected, which variables were analyzed, or how the statistical analyses were performed. Additionally, inclusion or exclusion criteria for patients are not mentioned, which is crucial in a study of this kind.

Incoherent presentation of results: The results mentioned, such as the reduction in operative time and the decrease in complications and infections, seem out of place in the methods section. Additionally, the results should be presented in more detail, providing specific effect measures, confidence intervals, and p-values.

Weak conclusions: While it concludes that thoracic surgery nurses require more specialized training, this conclusion appears to be a general deduction and is not supported by the specific findings of the study. Furthermore, there is no discussion of how the study's findings contribute to this conclusion.

In summary, the abstract requires significant revision to improve the clarity of the objectives, provide a more detailed description of the methodology used, present the results coherently, and adequately support the conclusions with the study's findings.

It would be beneficial for the introduction to provide more context on the importance of this topic and why it is relevant to investigate how nursing experiences affect patients in this specific context. This would help readers better understand the motivation behind the study and its clinical significance.

Additionally, the introduction could benefit from a brief review of relevant literature to further contextualize the study and demonstrate how it integrates into the existing body of knowledge on the subject. This could include previous studies that have investigated the influence of nursing experiences on surgical outcomes, particularly in patients with onco-hematological conditions.

This article does not present a scientific methodology or study variables.

The section lacks a clear description of the methodology used to conduct the study. There is no information provided on the study design, participant selection criteria, data collection procedures, or the statistical analysis used. This lack of detail makes it difficult to understand how the research was conducted and the reliability of the results.

The explicitly mentioned study variables that were evaluated during the research are not mentioned. The failure to identify these variables makes it difficult to understand which specific aspects of nursing work were being evaluated and how they relate to patient outcomes. The potential limitations of the study, such as selection bias, data validity, or any other factors that may have influenced the results, are not discussed. Identifying and discussing these limitations is important for properly interpreting the findings and understanding their applicability in other contexts.

this Materials and Methods section lacks the necessary details to support the validity and reliability of the study results. A thorough revision is suggested to provide a more complete and clear description of the methodology used, as well as explicit identification of study variables and a discussion of potential study limitations.

Based on what has been discussed so far and considering the conclusions, it is evident that the article lacks coherence and proper methodology. It needs to be completely restructured to ensure it meets scientific and research standards. Creating tables to visualize and understand the calculations performed is suggested, which would improve the clarity and presentation of the results. Additionally, while images may be suitable for a conference presentation, they are not appropriate for a scientific article. It is important to review and modify the structure of the article to ensure it is coherent, clear, and follows the principles of scientific research.

Reviewer 3 Report

Comments and Suggestions for Authors

The topic is really interesting and brings value to nursing care and its professionals. The introduction lacks some important topics for contextualizing the topic and justifying its interest. the objective should be revised to be more congruent with the topic and should be the same in the different parts of the paper (abstract, introduction and method).

The methodology is very incomplete in describing the type of study, the instruments used, the definition of the two groups (DNT and Nom-DNT) and ethical considerations and procedures,It is essential to be described here. There are aspects that should be included in the introduction as a justification and contextualization of the problem and presented here in another format (observation grids and document analysis). Images are not necessary. The findings were interesting, significant in relation to the objectives, and the discussion of the data was well done, with a comparison of authors who can be updated with more recent studies. They should look for more current studies, given the topic under study, they should be after 2014. It was interest, given the objectives of the study, to be able to divide the discussion into two topics. The conclusions are concise and focused on the study's central objective, references should not be used here, as at this point the main results found are summarized and the main conclusions of this study are presented. The references require updating, especially those prior to 2014.

Reviewer 4 Report

Comments and Suggestions for Authors

Dear authors,

The choice of the subject is very good and interesting; however, there are several problems in all sections of the article. Τhe most serious issue that you need to work on is plagiarism (48%). I hope you find my comments helpful. In detail, my comments are as follows:

Title

Capitalize the first letter of every word in the title.

Abstract

The abstract should follow the style of structured abstracts, but without headings.

Introduction

-        - The introduction is too brief and unconvincing to both identify the gap in the literature that this study seeks to fill and to outline and assess the fundamental issue and body of scientific knowledge. Which particular research gap is addressed in this paper?

- The introduction should highlight the study's importance and provide a brief explanation of the study's overall background. When compared to other published content, what does it bring to the topic area? Examining the condition of the field's research in detail and mentioning any significant works is crucial. When necessary, please draw attention to contradictory and controversial findings from scientific studies.

Materials and Methods

-        Line 72: Give a clearer description of the methodology. Ιn this period of time what is the DNT period and what is the non-DNT period?

-        Lines 74-76: Τherefore the same nurses were included in the non DNT period and then after training were considered as DNT period; The process is not clear. If they are the same nurses, a more specific description of their joint training would need to be provided to be considered as DNT and how long the training lasted.

-        Line 81: “….for a period of at least nine months”. Τherefore a length of service in this department of less than 9 months was a criterion for exclusion from the study? Add to the criteria described above.

-        Lines 11 & 149: Please dot after the brackets (figure 4).

-        Lines 352-362: Please include in the text of materials and methods the description of the procedure for ethics committee approval and informed consent. Moreover, regarding figures included images of patients, written informed consent for publication must be obtained from participating patients who can be identified (including by the patients themselves). Please state “Written informed consent has been obtained from the patient(s) to publish this paper” if applicable.

-        How did you address the aim of the study as it is described in Introduction section (lines 62-67: The aim of the present study is to demonstrate with a team of specialized nurses the fundamental role in decreasing perioperative stress, the duration of operations, minimizing postoperative complications and reducing hospital stay times, improve the coordination of the work of surgeons and anesthetists, and evaluate the effects on work efficiency and patient outcomes in thoracic surgery). `

Results

-        Line 167: The demographics of nurses should also be presented in table. The same question about the methodology remains. Were these nurses in the DNT period? Were they the same in the non-DNT period that were then trained; if they were other nurses in the non-DNT period, a description of their demographics is needed as well.

-        Lines 175-212: The tables of these results are missing.

-        Line 182: please erase (table 1) since you have already start the sentence with the number of the number.

-        Line 188: Τhe tables are not a separate subsection. Τhe same applies to the other subsections. Also, what is subsection 3.1?

-        Lines 194-195: Please explain the procedure.

-        Lines 196-204: Ηow does the statistical significance of these results emerge? add the corresponding table.

-        Table 2 and in the text: Please correct p=0.00001 as p<0.001.

-        Line 222: Please dot after the brackets (table 2). Make the corrections in all parts of the text.

-        The reduction of perioperative stress, which is stated as the purpose of the study, is not shown anywhere in the results.

Discussion

-        The results and how they might be interpreted in light of earlier research and working hypotheses should be reported by the authors. Results and this section could be combined. This discussion mostly consists of generalizations; while it falls short of comparing the results of this study with the body of international literature

-        Future directions for research can also be addressed. Future research directions may also be mentioned.

Conclusions

-        Please explain how the evidence and arguments offered support the findings or do not support them. Please specify which specific experiments were used to address each of the primary concerns that were raised.

References

Please correct the reference list according to the instructions for authors. For example: Abbreviated Journal Name YearVolume, page range.

Comments on the Quality of English Language

Minor editing of english language is required

Reviewer 5 Report

Comments and Suggestions for Authors

my main suggestion is to review the attached documents

as there are major changes that need to  be considered

there should be a process to make sure the basic rules of publishing are followed

the authors are encouraged to review the research process.

see above comments for the authors plus the following:

Additional comments

INTRODUCTION

The introduction is confusing as many concepts are introduced but the connection, relevance are not appropriate. Important to really focus on the issue that you identified which seems to be highly qualified OR nurses but why?

MATERIALS AND METHODS

Why did  you choose a retrospective design as no information is provided int the introduction/background?

The inclusion criteria are vague?

 What do you mean by “whom data regarding indictors of the quality of nursing?”

What about what you mean by 2 periods, patients, nurses ? DNT, non DNT

The exclusion criteria for both patients and nurses?

This section is about materials and methods, and you did put a lot of emphasis in the following:

-          Roles and steps of nurses (Is it linked to DNT?)

-          Description of the surgical procedures

You need to increase the discussion about this DTN vs non DTN

How did you extract the data about all the outcomes collected which you should have provided the rationale for using them as previously mentioned inside the article.

The methods have no rationale and no research question only the aim in the introduction but ? link with the gap;  you should have identified and added but it is not available.

Why so many criteria to determine the difference between 2 periods which are certainly obscure- need to add what exactly what they are…

Comments on the Quality of English Language

some minor typos and some sentences which are incomplete.
